# Endocrine modulation of primary chemosensory neurons regulates *Drosophila* courtship behavior

Matthew R. Meiselman [1,2]*, Anindya Ganguly [2,3], Anupama Dahanukar [1,2,3], Michael E. Adams [1,2,3,4]*

**1** Graduate Program in Cell, Molecular and Developmental Biology, University of California, Riverside, California, United States of America, **2** Department of Molecular, Cell, and Systems Biology, University of California, Riverside, California, United States of America, **3** Graduate Program in Neuroscience, University of California, Riverside, California, United States of America, **4** Department of Entomology, University of California, Riverside, California, United States of America

* matthew.meiselman@UNLV.edu (MRM); adams@ucr.edu (MEA)

## Abstract

The decision to engage in courtship depends on external cues from potential mates and internal cues related to maturation, health, and experience. Hormones allow for coordinated conveyance of such information to peripheral tissues. Here, we show Ecdysis-Triggering Hormone (ETH) is critical for courtship inhibition after completion of copulation in *Drosophila melanogaster*. ETH deficiency relieves post-copulation courtship inhibition (PCCI) and increases male-male courtship. ETH appears to modulate perception and attractiveness of potential mates by direct action on primary chemosensory neurons. Knockdown of ETH receptor (ETHR) expression in GR32A-expressing neurons leads to reduced ligand sensitivity and elevated male-male courtship. We find *OR67D* also is critical for normal levels of PCCI after mating. ETHR knockdown in OR67D-expressing neurons or GR32A-expressing neurons relieves PCCI. Finally, ETHR silencing in the corpus allatum (CA), the sole source of juvenile hormone, also relieves PCCI; treatment with the juvenile hormone analog methoprene partially restores normal post-mating behavior. We find that ETH, a stress-sensitive reproductive hormone, appears to coordinate multiple sensory modalities to guide *Drosophila* male courtship behaviors, especially after mating.

## Author summary

The decision of when to reproduce is paramount for organismal survival. In models like mice and flies, we have a comprehensive understanding of neuronal substrates for perception of mates and courtship drive, but how these substrates adapt to malleable internal and external environments remains unclear. Here, we show that post-mating refractoriness depends upon a peptide hormone, Ecdysis-Triggering Hormone (ETH). We show repression of courtship toward recently-mated females depends upon pheromone cues and that ETH deficiency impairs perception of female matedness. ETH signaling appears to promote the activity and function of pheromone-sensing primary olfactory and

**Data Availability Statement:** All relevant data are within the manuscript and its Supporting Information files. Further information and requests

for resources and reagents should be directed to and will be fulfilled by the corresponding authors, Matthew Meiselman (matthew.meiselman@UNLV.edu), and Michael E. Adams (adams@ucr.edu). This study did not generate new unique reagents. All reagents used in the study are available upon request.

**Funding:** This work was supported by the National Institutes of Health (NIH) GM-067310 (to MEA), the UCR Agricultural Experiment Station, The UCR Office of Research, and the UCR Graduate Division. Facilities provided by the UCR Institute for Integrated Genome Biology were instrumental to the success of this study. The funders had no role in study design, data collection and analysis, decision to publish, or preparation of the manuscript.

**Competing interests:** The authors have declared that no competing interests exist.

gustatory sensory neurons. Additionally, ETH sets internal levels of Juvenile Hormone, a hormone known to inhibit courtship drive in flies. Elimination of ETH or its receptor in primary sensory neurons or the glandular source of Juvenile Hormone reduces male post-copulation courtship inhibition (PCCI), causing continued courtship toward female counterparts after successful mating. Our data suggest ETH and its targets are critical for post-mating refractoriness in males.

## Introduction

Modulation of sensory perception is critical for prioritization of appropriate behaviors under varying physiological conditions. For example, in the fruit fly *Drosophila melanogaster*, mating is costly for both sexes [1,2], and the decision to engage in mating behaviors must be weighed with internal state and probability of success [3,4]. *Drosophila* males weigh these decisions with a network of neurons masculinized by the presence of the sex determination transcription factor, *fruitless* [5]. Reproductive decision-making and consequent behavioral output in male flies are dictated by the activity of P1 neurons [6,7]. These neurons integrate olfactory, gustatory, visual, and hormonal cues to determine the levels of arousal and the ultimate decision to engage in courtship activity [8–10]. In addition to direct modulation of the arousal hub, it appears that primary sensory neurons necessary for mate perception also are modulated by hormonal state [11]. This layered control allows for multifactorial integration into the ultimate decision of whether or not to mate.

We recently reported that Ecdysis Triggering Hormone (ETH) persists into the adult stage where it modulates juvenile hormone levels, fecundity and reproductive physiology in response to stress [12,13]. Here, we characterize postcopulation courtship inhibition (PCCI) in males and show that blocking ETH release or elimination of Inka cells, the sole source of ETH, relieves (PCCI) and increases male courtship overtures toward conspecific females and males. Inappropriate courtship toward males and mated females after mating suggests ETH-deficiency obviates not only the variety of courtship-inhibiting olfactory and gustatory cues that distinguish each [14–16], but also any changes in memory or internal state that result from successful copulation [17]. Furthermore, ETH receptor (ETHR) knockdown in aversive phero-mone-sensing neurons that express *GR32A* and *OR67D* relieves PCCI and promotes male-male courtship, suggesting that ETH signaling modulates courtship behavior by affecting pher-omone sensitivity. Juvenile hormone (JH) deficiency also relieves PCCI; treatment of ETH-deficient males with the juvenile hormone analog methoprene (JHA) reestablishes a decline in courtship intensity after mating. We conclude that ETH, a stress-sensitive reproductive hormone, is a potent modulator of courtship, and necessary for normal levels of PCCI.

## Results

### ETH deficiency disinhibits post-copulation male courtship behavior

Normal levels of male courtship behavior depend upon JH signaling [18]. We therefore examined whether males deficient in ETH, which stimulates synthesis and release of JH [12,19,20], exhibit impaired mating efficacy. Contrary to expectation, we found that ETH deficiency, caused either by expression of the pro-apoptotic protein *reaper* in the Inka cells during the adult-stage (*ETH-Gal4;Tubulin-Gal80^{ts}>UAS-Reaper*) [21], or by adult-specific prevention of ETH release by overexpression of the temperature-sensitive version of the dynamin mutant, *shibire^{ts}* (*ETH-Gal4>UAS-shibire^{ts}*) [22], increases male-female copulation success during a

ten-minute test interval (Fig 1A and 1B). Furthermore, injection of ETH into wild-type *Canton-S* males just before pairing with a virgin female dramatically attenuates courtship (Fig 1C). Additionally, both Inka cell-ablation and Inka cell secretion-blocking cause a substantial increase in courtship toward conspecific males (Fig 1D and 1E, S1A and S1C Fig). Most strikingly, we found that ETH-deficient male courtship toward females does not decrease after successful mating, as was observed in wild-type flies and genetic control flies (Fig 1F and 1G, S1B, S1D and S1E Fig). We refer henceforth to this decrease as "post-copulatory courtship inhibition" or PCCI. Elevated courtship after mating also was observed in males with ecdysone receptor knockdown in Inka cells (*ETH-Gal4;Tubulin-Gal80ts>UAS-EcR-RNAi*), which reduces ETH gene expression [23], suggesting the phenomenon is ETH-dependent (S1F Fig). This is unlikely to be a consequence of incomplete mating, as sperm were observed in females after mating to ETH-deficient males (S2A–S2C Fig), copulation duration for ETH-deficient males was slightly longer than controls (S2D and S2E Fig), and we showed in previous work that ETH-deficient males could successfully produce progeny, albeit at a reduced rate [12].

Male flies typically reduce courtship overtures after mating [15]. This reduction depends upon internal cues from copulation reporting neurons, which suppress courtship-promoting NPF neurons [3,24], and sensing of external cues that signal female refractoriness [25–27]. Males transfer pheromones, including cis-Vaccenyl Acetate (cVA) and 7-tricosene (7T) to females during copulation. This contributes to male aversive behavior during subsequent encounters with females [28]. ETH-deficient males may therefore either be impaired as "senders" (inability to deposit anti-aphrodisiacs) or "receivers" (inability to perceive or integrate information from anti-aphrodisiacs). To clarify, we performed a "Mate-and-Switch" experiment by pairing ETH-deficient and wild-type males with females simultaneously, and swapping their former mates after completion of mating (Fig 2A). *Canton-S* (control) males court females formerly mated to ETH-deficient males at low levels, whereas ETH-deficient males courted mated females at high levels (Fig 2B). These findings suggest that ETH modulates the internal state of receiver males to tune courtship. Taken together, ETH levels appear to be critical for maintaining normal courtship behavior after mating.

## ETHR-silencing impairs GR32A neuron function and elevates male-male courtship

Males and mated females rely upon pheromones to communicate to other males that they are unsuitable as prospective mates [29]. We recently showed that suppression of male-male courtship depends upon ETH signaling in antennal lobe interneurons and that JH levels and ETH-JH signaling do not impact male-male courtship [30]. Since ETH-deficient males court conspecific males with more than twice the intensity reported from interneuron-specific ETHR-silencing, we manipulated neuronal ETHR expression to determine the extent of such ETH modulation. First, we knocked down ETHR pan-neuronally (*Elav-Gal4>UAS-ETHR-RNAi*), and observed a significant elevation of post-copulation courtship behavior (Fig 2C). To rule out developmental defects, we performed the same experiment with Tubulin-Gal80ts (*Elav-Gal4;Tubulin-Gal80ts*) and raised the flies at the restrictive temperature (18°C) until eclosion, followed by a shift to the permissive temperature (29°C) until Day 4, when PCCI was assessed (Fig 2D). Still, courtship after completion of mating was elevated. While ETH-Gal4 expression is largely limited to the epitracheal Inka cells in both sexes (S3A–S3D Fig) [12], a high-fidelity *ETHR-Gal4* [31] expresses in cells the brain, including Kenyon cells, peptidergic neurons, and olfactory neurons (Fig 3A–3D) [32]. We compared expression of *ETHR-Gal4* in male and female nervous systems, including the sensory periphery (Fig 3). Intriguingly, we found *ETHR-Gal4* labels olfactory and gustatory sensory neurons, which are

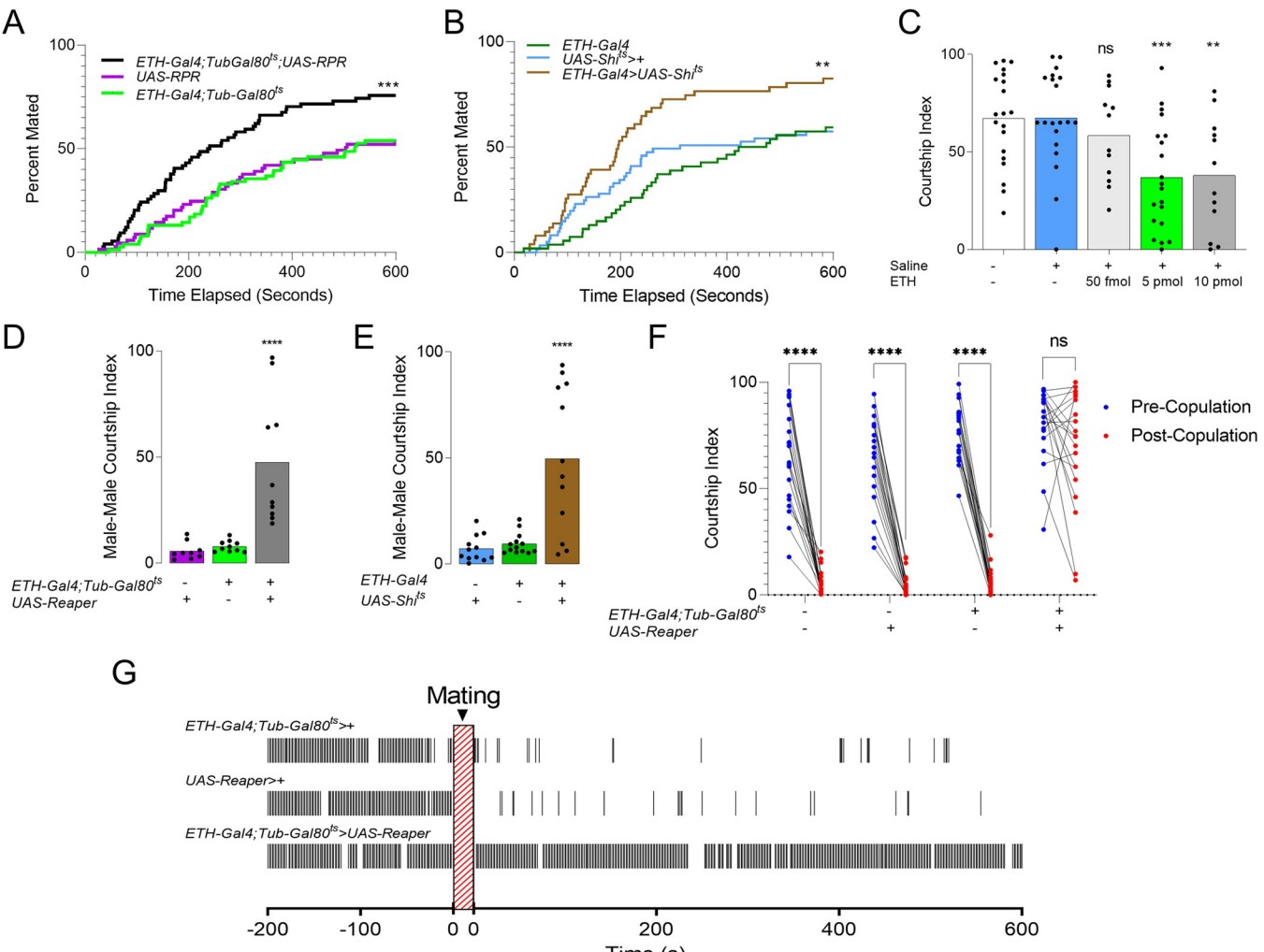

**Fig 1. ETH-deficiency increases frequency of male courtship behavior toward females, males, and mated females.** (**A-B**) Cumulative totals of males mating successfully with virgin *Canton-S* females during a ten-minute pairing interval. Male genotypes are: (**A**) Inka cell-ablated, (*ETH-Gal4;Tubulin-Gal80$^{ts}$>UAS-Reaper*)(Mantel-Cox test, n = 70–77), (**B**) Inka cell secretion-blocked, (*ETH-Gal4>UAS-Shibire$^{ts}$*) and genetic controls (Mantel-Cox test, n = 51–61). (**C**) Male courtship index toward *wt* females beginning 45 minutes after faux injection (no liquid ejected from capillary), saline injected, or injected with stated amount of ETH in saline (ANOVA, n = 12–20). (**D-E**) Male courtship index (time spent courting over total time, 600s) toward *Canton-S* males for Inka cell-ablated (**D**), Inka cell secretion-blocked (**E**), and genetic controls (One-way analysis of variance (ANOVA), n = 15–20). (**F**) Comparison of courtship index before (Blue dots) and after (Red dots) successful mating for *Wt* (left), controls (middle) and Inka cell-ablated males (right)(bootstrap, n = 20) (**G**) Time courting 200 seconds before mating and 600 seconds after dismount for an example Inka cell-ablated male and genetic controls. Black bars represent seconds performing courtship behavior. ns $p$ > .05; ** $p$ < .01; *** $p$ < .001; **** $p$ < .0001.

used for mate discrimination (Fig 3B, 3D, 3G–3J) [33]. We identified several notable differences in expression, with female-specific labeling of neurons in the pars intercerebralis (Fig 3A and 3C), male-specific labeling of neurons in the mesothoracic and abdominal neuromeres (Fig 3E and 3F) and the prothoracic tarsi (Fig 3G and 3H). While the antennal lobe was similar in both sexes, we identified neurons innervating VA1lm and DA1 glomeruli, which are post-synaptic targets of olfactory sensory neurons that express OR47B and OR67D, respectively (Fig 3B and 3D) [34]. These sensory neuron subsets are *Fru$^{M}$*-positive olfactory neurons that convey pheromone information to the brain to tune male courtship behavior [35,36]. GR32A-expressing gustatory neurons also sense pheromones [37], and *ETHR*- and *GR32A*-driven

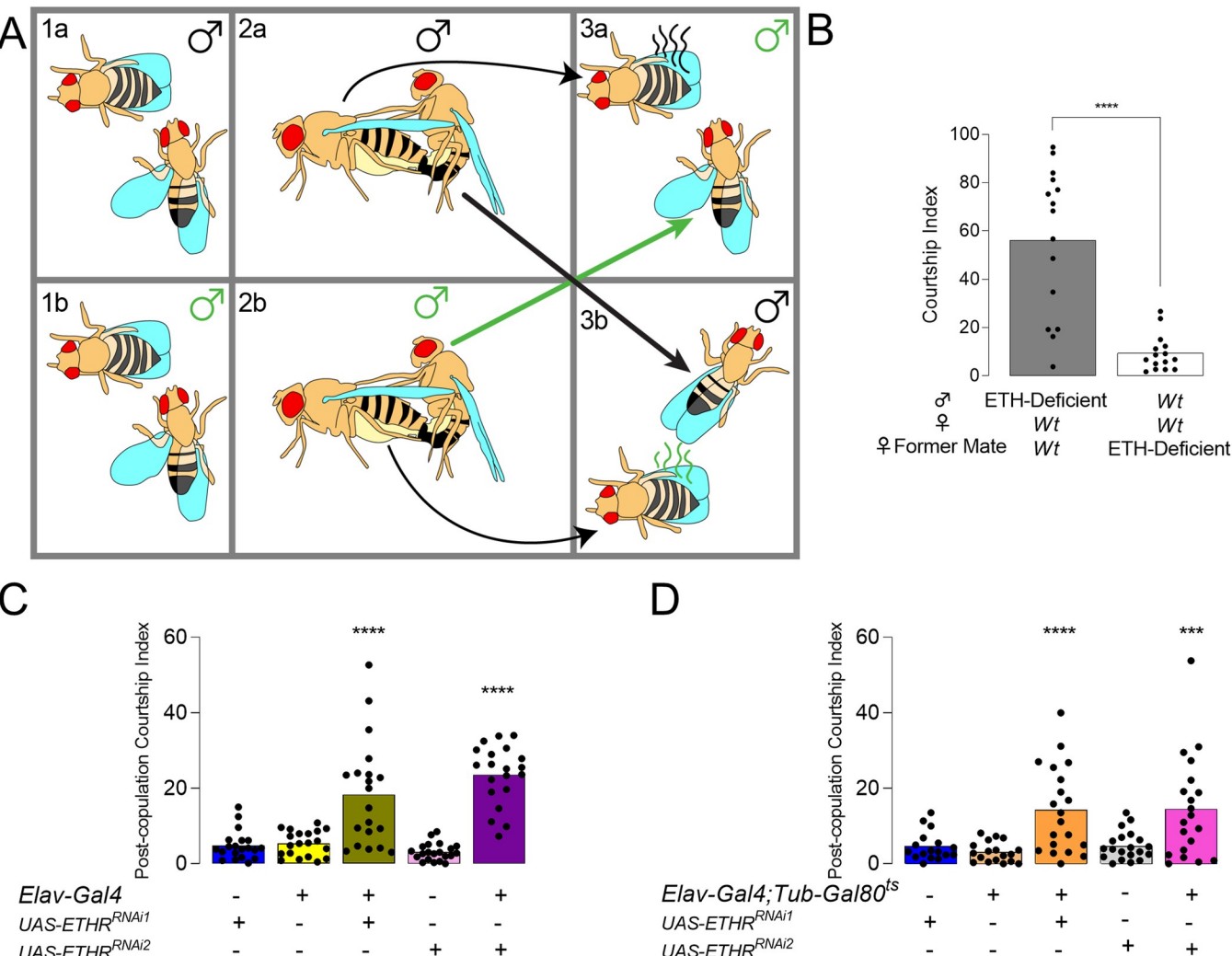

**Fig 2. ETH deficiency impairs perception.** (**A**) Diagram for Mate and Switch experiment. *Wt* and ETH-deficient males court and mate *wt* female counterparts (1a-2a, 1b-2b, respectively). After completion of mating, ETH-deficient males are placed with females mated to *wt* male counterparts (3a), and *wt* males are placed with females recently mated to ETH-deficient males from the opposite panel (3b). (**B**) Courtship index after mate and switch for ETH-Deficient and *wt* males after mating and switching (Mann-Whitney U, n = 15). (**C**) Male post-copulation courtship index for flies with ETHR knocked down pan-neuronally (*Elav-Gal4>UAS-ETHR-RNAi*) and genetic controls (ANOVA, n = 20–21).(**D**) Male post-copulation courtship index for flies with ETHR knocked down pan-neuronally after eclosion only (*Elav-Gal4;Tubulin-Gal80^ts^>UAS-ETHR-RNAi*) and genetic controls (ANOVA, n = 17–20).ns, $p >$ .05; *** $p < .001$; **** $p < .0001$.

fluorophore expression (*ETHR-Gal4;UAS-mCD8-GFP>GR32A-lexA;AOPmCherry*) appears to overlap in cells in the labellum, but not in the tarsi (S4A and S4B Fig).

We therefore hypothesized sensory neurons may be a target of ETH and assessed candidate subsets known for sensing male-derived, courtship-inhibiting volatiles GR32A, OR47B and OR67D [37–39]. As ETH-deficient males court other males intensely, we assessed the requirement of ETHR expression in each of these classes of neurons for inhibition of male-male courtship (Fig 4A). Males subjected to *ETHR* knockdown in GR32A neurons, but not in OR47B+ or OR67D+ neurons, significantly increased courtship toward conspecific males, suggesting that ETH directly influences GR32A+ neuron sensitivity.

The overlap in labellar but not tarsal expression of *GR32A-LexA* and *ETHR-Gal4* suggests ETH may target pheromone-sensing neurons in the labellum. GR32A+ neurons in the

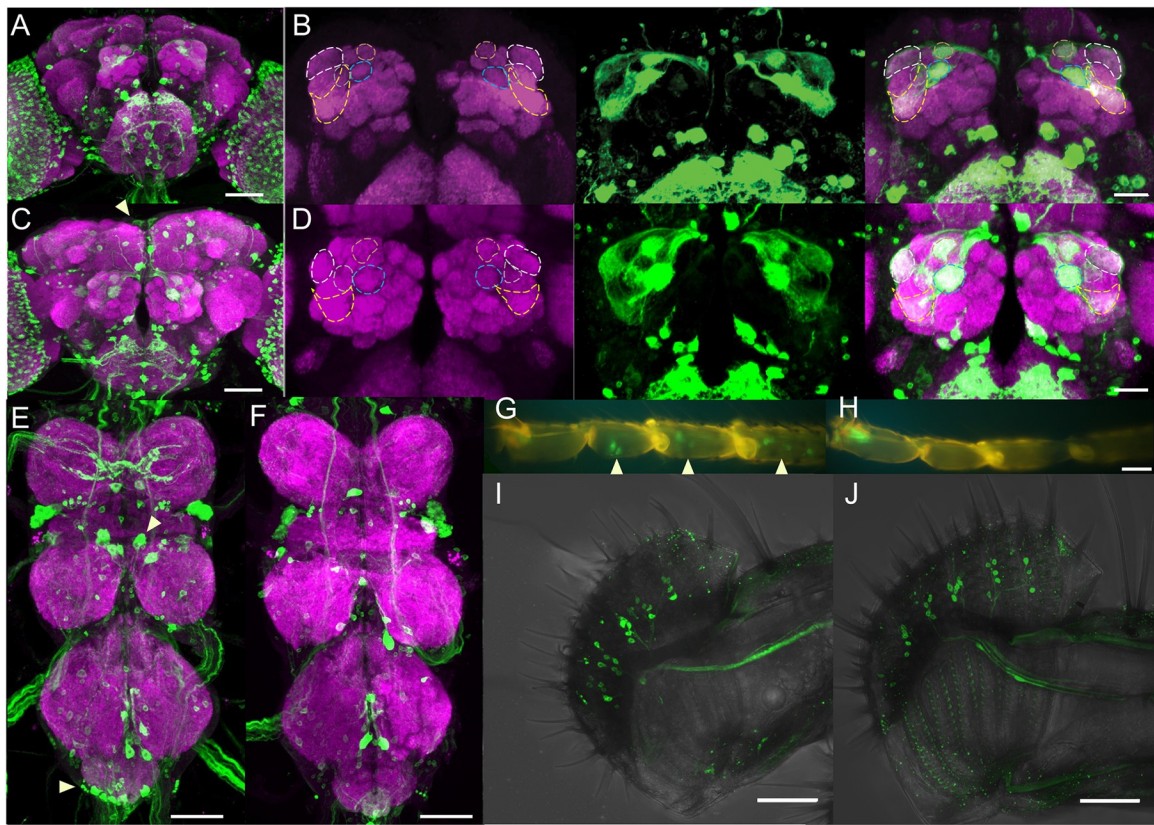

**Fig 3. Sexually dimorphic expression pattern of *ETHR-Gal4* in male and female nervous systems. (A-D)** Brains of example *ETHR-Gal4>UAS-mCD8-GFP* male (**A-B**) and female (**C-D**)(green is anti-GFP, magenta is anti-bruchpilot, scale bar = 50 μm). Antennal lobes from **A** and **C** are enlarged in **B** (male) and **D** (female) with ETHR-Gal4-labeled glomeruli demarcated by dashed lines in left panel (DA1/OR67D-white, VA1l/m/OR47B-yellow, VA6/OR82A-blue, DA4m/OR2A-beige and DC3/OR83C-gray), GFP alone in middle, and merged in right panel (scale bar = 25 μm). Thoracic ganglia of example *ETHR-Gal4>UAS-mCD8-GFP* male (**E**) and female (**F**)(scale bar = 50 μm). **(G-J)** *ETHR-Gal4>UAS-mCD8-GFP* expression in the taste-sensing periphery of male (**G**, **I**) and female (**H**, **J**) prothoracic tarsi (**G-H**) and labella (**I-J**) (scale bar = 50 μm). Yellow arrowheads in **C** (pars intercerebralis), **E** (mesothoracic ganglion), and **G** (tarsus) indicate cell bodies of neurons only observed in one sex.

labellum are activated by 7T as well as a variety of other bitter tastants [40,41]. We tested responses to denatonium in ETHR-knockdown (*GR32A-Gal4>ETHR-RNAi*) and control males, and found a small, but significant reduction in action potentials with a slightly longer latency to response over a range of concentrations (Fig 4B and 4C, S4C and S4D Fig). Finally, we tested whether GR32A-driven *ETHR* knockdown impacted spiking after contact with 7T. ETHR knockdown decreased responses to 5 mg/ml 7T, which may explain observed elevation in male-male courtship upon ETHR silencing in GR32A neurons (Fig 4D and 4E). Thus, ETHR knockdown results in diminished sensitivity to ligands in primary sensory neurons, including pheromone.

## ETHR silencing in OR67D and GR32A-expressing neurons relieves post-copulatory courtship inhibition (PCCI).

As mentioned above, males "perfume" females with cVA and 7T during mating, which communicate their mated status to future male suitors [27]. These pheromones are sensed by OR67D-olfactory and GR32A-gustatory neurons, respectively, two mutually exclusive sensory neuron populations whose activity inhibits courtship. We investigated whether these pathways

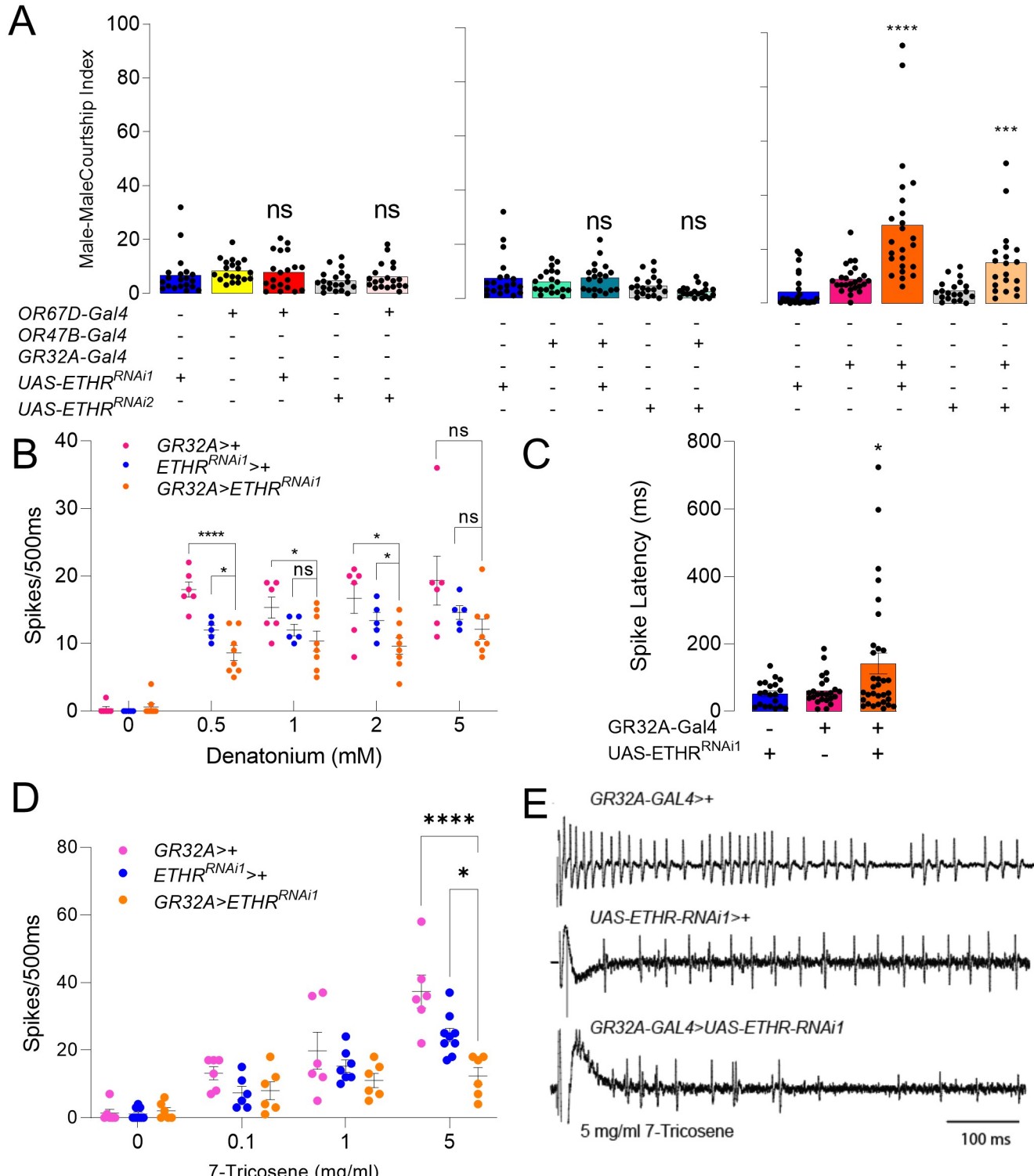

**Fig 4.** (**A**) Male-male courtship index of males with ETHR knockdown in pheromone sensory neurons OR67D, OR47B, and GR32A with genetic controls (ANOVA, n = 15–25). (**B**) Extracellular recordings from S6 taste hairs of *GR32A-Gal4>UAS-ETHR^{RNAi1}* and control males to 0, 0.5, 1, 2, and 5 mM denatonium for

500 ms after contact (2-way ANOVA with Tukey's test for multiple comparisons, n = 6–8). (**C**) Latency to first spike after denatonium exposure for *GR32A-Gal4>UAS-ETHR$^{RNAi1}$* and controls across all concentrations (Pairwise One-way ANOVA). (**D**) Mean neuronal responses obtained for the first 500 ms from s6 sensilla of the *GR32A-Gal4>UAS-ETHR-Sym, UAS-ETHR$^{RNAi1}$*, and *GR32A-Gal4>+* males upon stimulation with the indicated concentrations of 7-T. (Each dot represents responses obtained from one individual/taste sensillum. 2-way ANOVA with Tukey's test for multiple comparisons, n = 6–9). (**E**) Representative traces obtained for the first 500 ms from s6 sensilla of the *GR32A-Gal4>UAS-ETHR$^{RNAi1}$*, *UAS-ETHR$^{RNAi1}$>+*, and *GR32A-Gal4>+* males upon stimulation with 5 mg/ml 7T (scale bar = 100 ms). ns $p > .05$; * $p < .05$; ** $p < 0.01$; *** $p < 0.001$; **** $p < .0001$.

are important for inhibition of courtship immediately after mating. To eliminate the *GR32A* gene in sensory neurons, we used the minos-mediated integration cassette *GR32A$^{mimic}$*, which has GFP inserted into the *GR32A* locus [42], and knocked down GFP with a pan-neuronal driver, *57C10-Gal4* [43]. This technique allows for specific and extremely effective knockdown of tagged genes [44]. We first assessed whether knockdown of GR32A de-repressed male-male courtship as was previously reported [45]. As expected, GR32A knockdown stimulated male-male courtship (Fig 5A). Knockdown also diminished PCCI, arguing the behavior relies upon *GR32A* and 7T (Fig 5B). We observed high post-copulation courtship in controls, which we attribute to males being heterozygous for native *GR32A*. *OR67D$^{-/-}$* knock-out mutants also had impaired PCCI (Fig 5C). Next, we knocked down ETHR in OR67D, OR47B and GR32A-expressing neurons and found that ETHR-silencing in *OR67D* and *GR32A*-expressing subsets relieves PCCI (Fig 5D). These results suggest that ETH signaling modulates pheromone perception to tune male courtship both before and after mating.

## Juvenile hormone regulates post-copulation courtship

While reduced ligand sensitivity can account for elevated male courtship toward males and mated females, ETH-injected flies reduce courtship toward virgin females (Fig 1C), which do not have significant levels of cVA or 7T. ETH-deficient males have low JH levels [12,46], and JH inhibits the activity of dopaminergic, pCd, and NPF neurons upstream of P1 courtship command neurons, thus preventing courtship [47]. We therefore sought to determine if JH plays a role in the disinhibition of post-copulation courtship observed in ETH-deficient males. We knocked down the ETH receptor (ETHR) specifically in the CA, which reduces total JH levels (*JHAMT-Gal4<UAS-ETHR-RNAi*) [12]. We observed a significant elevation in post-copulation courtship, suggesting JH may also play a role in post-copulation courtship (Fig 5E). Accordingly, we tested whether treatment with the JH analog, methoprene, could restore normal post-mating refractoriness (Fig 5F). Methoprene treated, ETH-deficient males significantly decreased courtship after mating, but courtship after mating was still elevated compared to controls (Fig 5F). Thus, JH appears to be critical for post-copulation courtship inhibition, but insufficient to completely account for elevated post-copulation courtship in ETH-deficient males. We therefore propose that ETH inhibits courtship indirectly by promoting JH synthesis and directly by setting sensitivity to courtship-inhibitory pheromones to normal levels (Fig 5G).

## Discussion

We show here that the peptide hormone ETH is necessary for courtship discrimination and post-copulation courtship inhibition. ETH-deficient flies show elevated courtship intensity toward same and opposite sex conspecifics and, strikingly, ETH-deficient males do not terminate courtship overtures toward females after successful copulation. Furthermore, we found that ETH-deficient males are impaired in pheromone sensing or processing, and injection of ETH attenuates courtship intensity toward females. signaling in *GR32A*-expressing neurons is necessary for normal responses to ligands and prevents courtship toward other males. Post-copulation courtship is inhibited by *OR67D*, and ETHR-silencing in *OR67D* and *GR32A*-

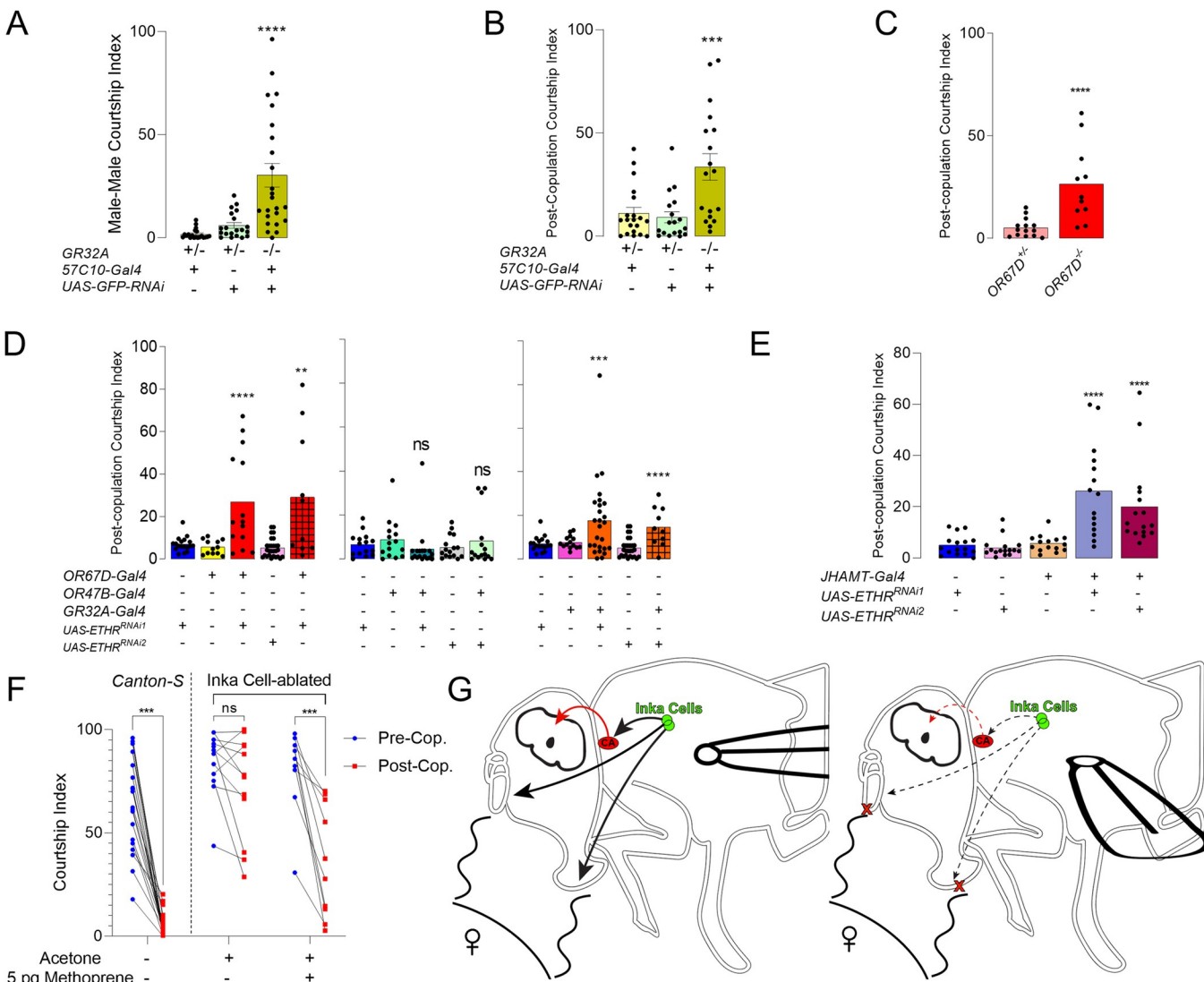

**Fig 5. Post-copulation courtship inhibition depends upon *GR32A*, *OR67D*, ETHR, and Juvenile Hormone.** (**A**) Quantification of male courtship toward a *wt* male subject for *GR32A*[mimic] mutants and heterozygous genetic controls (ANOVA, n = 20–24). (**B**) Quantification of male courtship toward a *wt* female subject after copulation for *GR32A*[mimic] mutants and heterozygous genetic controls (ANOVA, n = 18–20). (**C**) Post-copulation courtship for *OR67D*[-/-] males and heterozygous controls (Mann-Whitney U, n = 10–18). (**D**) Post-copulation courtship index for males with ETHR knockdown in OR67D, OR47B, and GR32A neurons and genetic controls (ANOVA, n = 10–25). (**E**) Post-copulation Courtship for *JHAMT-Gal4>UAS-ETHR-RNAi* flies and controls. (**F**) Pre-copulation and post-copulation courtship for reference *wt* males, and Inka cell-ablated (*ETH-Gal4;Tubulin-Gal80^{ts}>UAS-Reaper*) males treated with either acetone alone or acetone with the Juvenile Hormone Analog, methoprene (n = 10–12). (**E**) Model for the role of ETH in modulation of courtship. In normal conditions (Left), Inka cells (green) release ETH (black arrows), stimulating the CA to produce JH (Red, Red arrow) for subsequent inhibition of P1 circuitry, and promote activity in aversive pheromone (wavy lines) sensing neurons in the antennae (OR67D⁺) and labellum (GR32A⁺). This results in low courtship (wings at side). When ETH is low (indicated by black dashed arrows), the CA and aversive pheromone-sensing neurons are less active, resulting in lower JH (dashed red arrows) and insensitivity to aversive pheromones (red x's), thus stimulating courtship and unilateral wing extensions (wing extended outward). ns $p>0.05$; ** $p < .01$; *** $p < 0.001$; **** $p < .0001$.

expressing neurons relieves post-copulation courtship inhibition. Silencing ETHR in the CA also stimulates post-copulation courtship, while treatment of JHA to ETH-deficient males restores a measure of postcopulatory courtship inhibition after mating.

Our findings support a model whereby normal courtship behavior of males depends upon circulating ETH levels in the hemolymph. At low levels of ETH or ETHR expression, males are less responsive to courtship inhibitory pheromones with respect to both primary

chemosensory neuron responsiveness and behavior. JH inhibits courtship in early adulthood [47], and ETH levels are tightly linked with JH. It is therefore likely that ETH targets multiple, semi-redundant signaling systems (GR32A, OR67D, CA) to tune courtship levels rapidly in response to stimuli. Indeed, ETH is responsible for regulation of reproduction in response to stress. Ecdysone levels also fluctuate during courtship [48], and directly target OR67D neurons to regulate male courtship [49]. Interestingly, synthesis and release of ETH is tightly regulated by ecdysone, and ETH shares courtship-regulating targets with both upstream ecdysone (OR67D), and downstream JH (OR47B) signals; these three hormones may work together to co-regulate timing of courtship drive, though this requires further investigation.

## ETH signaling in primary sensory neurons

ETH modulates a variety of courtship targets and appears to have a major influence over the drive to mate. Our previous report showed ETHR expression in antennal lobe interneurons is important for mate discrimination [30]. Elimination of ETHR in these interneurons causes disinhibition of male-male courtship, possibly through cVA desensitization [50,51]. We show here that knockdown of ETHR in OR67D neurons regulates post-copulation courtship, which is likely to be cVA-dependent. ETHR silencing in either component of the olfactory system disinhibits normal courtship, suggesting specific physiological roles for ETH in male courtship behavior.

ETHR is a Gq-coupled GPCR, stimulating calcium release from intracellular stores upon activation [52]. During the ecdysis sequence, peptidergic ensembles of ETHR neurons mobilize calcium and release peptide signaling molecules that activate centrally patterned behaviors [53]. In this study, excitability of ETHR-expressing chemosensory neurons appears to be modulated by circulating ETH, becoming more or less sensitive to ligands as hormone levels fluctuate, as we observed in GR32A labellar neurons. It is currently unclear over what time frame such hormonal modulation in this context may occur. However, considering previously demonstrated ETH dynamics during ecdysis, the correspondence between ETH release and behavioral outcomes is expected to be a matter of minutes [53,54].

## ETH-JH signaling supports post-copulation courtship inhibition

We recently demonstrated that ETH signaling deficiencies decrease whole body juvenile hormone levels in both males and females [12,46]. In this work, we show that ETH deficiency or ETHR knockdown in the CA elevates courtship after mating. JH has been linked to male reproductive maturation, while JH receptors in OR47B neurons promote their receptivity to courtship stimulatory pheromones [11]. A recent report showed that JH suppresses courtship by mature males and inhibits calcium activity in courtship-promoting NPF neurons, doublesex-positive pCd neurons, and dopaminergic neurons [47]. JH levels do not influence male-male courtship [30], and post-copulation courtship does not seem ethologically beneficial at any life stage, arguing that this role for juvenile hormone is independent of homeostasis or development. Timing of circulating JH appears to be a critical for the courtship status quo, with too much or too little disrupting proper courtship behavior. In addition to its modulation of perception, JH targets Kenyon cells to mature and enhance memory [46,55], and drastically reduced JH levels could prevent flies from forming or accessing a "memory" of a recent, successful mating. Paired with sensory deficit, this may account for relief of post-copulation refractoriness observed here.

## Conclusion

We have established a relationship between ETH and courtship, wherein ETH maintains responsiveness in several courtship-inhibiting cellular targets (GR32A-expressing gustatory neurons, OR67D-expressing olfactory neurons and the CA). How endogenous ETH levels

may change and, in turn, manipulate male courtship awaits future study. A drop in ETH levels in response to stress would by extension elevate courtship toward conspecifics; though in many other species stress has an inhibitory effect on male libido [56–58].

We find the influence of ETH and JH in post-copulation courtship particularly intriguing, as the post-mating change in behavioral state has received less attention than pre-mating courtship. ETH appears to modulate courtship by targeting multiple levels of courtship circuitry, both by modulating sensory input to the P1 neurons (OR67D and GR32A) but also the motivation-driving proximal input to the P1 circuit through JH. Further investigation is needed to determine how the CRN circuit, which attenuates activity of P1 neurons to induce mating satiety, integrates hormonal changes. After mating, *Drosophila* females receive sex peptide, which is released slowly from sperm tails causing infradian elevation of JH [59–61], allowing females to accelerate oogenesis and replenish egg stores. The male must also replace accessory gland proteins after mating, which depend upon JH for their synthesis [62]. As males lack circulating sex peptide, this introduces the intriguing possibility that flies employ ETH, an innate allatotropin, to raise JH levels after copulation and inhibit post-copulation courtship. Indeed, in the closely-related male Caribbean fruit fly *Anastrepha suspensa*, JH levels increase after mating [63].

We have demonstrated here that courtship behavior is critically dependent upon ETH signaling, and suggest multiple targets for this dependency. Our findings raise further questions regarding how the now well-established neural circuitry underlying male courtship integrates endocrine state, particularly after mating.

## Materials and methods

### Fly strains

Flies used for immunohistochemistry were raised at 22˚C on standard cornmeal-agar media under a 12:12 hr light:dark regimen. All flies used for behavior experiments were backcrossed for at least five generations into the *Canton-S* background. Inka cell-ablated flies were raised at the Gal80ts permissive temperature (18˚C). When dark pupae appeared, male pupae were isolated and put in 96-well plates with food until eclosion to maintain naivety. Following eclosion, they were moved to culture tubes within a few hours and heated to the nonpermissive temperature (29˚C) for 24 hours, then moved to 22˚C until day 4. Inka cell-blocked flies were raised at 18˚C until eclosion and similarly isolated, but instead transferred to 29˚C until day 4. For both sets, cold controls were isolated similarly, but tubes were maintained in 18˚C until courtship assessment. All other genotypes were isolated in the same fashion but maintained at room temperature. Use of double-stranded RNA constructs for silencing of ETHR (UAS-ETHR-Sym (ETHR^RNAi1); UAS-ETHR-IR2 line (ETHR^RNAi2,VDRC transformant ID dna697) were described previously [64]. ETHR-Gal4 was obtained from Benjamin White (National Institute of Mental Health, Silver Spring). JHAMT-Gal4 was obtained from Brigitte Dauwalder [18]. The OR67D-Gal4 knock-in was obtained from Anandasankar Ray [39]. The remaining fly lines were obtained from the Bloomington Stock Center (Indiana University, Bloomington, IN): ETH-Gal4 (BL51982), Tubulin-Gal80ts (BL7017), UAS-Reaper (BL5824), UAS-Shibire^ts (BL44222), Canton-S (referred to as *Wt*, BL64349), Elav-Gal4 (BL8765), UAS-GFP (BL5137), OR47B-Gal4 (BL9983), GR32A-Gal4 (BL57622), UAS-EcR-RNAi (BL37059) GR32A-mimic (BL37451),UAS-GFP-dsRNA (BL44415), and 57C10-Gal4 (BL39171).

### Courtship assays

For all behavior experiments, males were isolated prior to eclosion in culture tubes with food. Naive day 4 adult male flies were placed in a 1 cm diameter courtship chamber with same-

aged wild type Canton-S male or virgin female subject and observed. All courtship scoring was performed single-blind.

*Male-Male Courtship*: To assess male-male courtship, 4-day-old test males were placed in 1 cm courtship chambers with virgin *Canton-S* males and time performing stereotypical courtship behaviors (Orientation, licking, tapping, chasing and wing extension) was totaled during a 10 minute window after a gate bisecting the chamber and separating the males was removed. Seconds engaging in courtship behaviors over the 600 second total was converted to a percentage termed courtship index (CI).

*Male-Female Courtship*: For male-female courtship, 4-day-old males were placed into 1 cm courtship chambers with *Canton-S* virgin females and recorded until 10 minutes after the final male from the last mating pair in the 14-chamber apparatus dismounted. For pairs that did mate, courtship index (see above) during the time to copulation was tabulated when relevant. The time where the male mounted the female to the time of dismount was recorded in all cases and reported where relevant. At the moment of dismount, recently-mated males were scored for courtship behaviors during the following 10 minutes, and courtship index was reported as with males, but labeled post-copulation courtship index to distinguish it in this context. For pre-copulation/post-copulation comparisons, courtship index (tabulated as a percentage of time performing stereotypical courtship behaviors as a percentage of total time) during the five minutes (or fewer if males mated earlier) before mating was compared to only the first five minutes after males and females broke copulation. Time between mounting and dismounting was recorded as copulation duration.

*Mate-and-switch*: Inka cell-ablated (*ETH-Gal4;Tubulin-Gal80^{ts}>UAS-Reaper*) and *Canton-S* males isolated as before at eclosion were isolated until day 4, and simultaneously placed in three identical courtship chambers with Canton-S virgin females simultaneously. An experimenter watched the chambers, and immediately after completion, aspirated the male and female out and moved the recently-mated male subject and female partner to a courtship chamber, either empty or containing a counterpart from a desired pair which had completed mating just prior to the transport pair. The new pair were on either side of a gate, and when the 14-well chamber was full of desired partners, the video recording began and gates were retracted. As 3 times as many pairs were being mated as were needed in the final chamber, the entire loading sequence was completed within 3 minutes after the first male dismounted. Conceptual schematic is displayed in Fig 2A.

## Immunohistochemistry

*Brain and Sensory Organ Staining*: We crossed *ETH-Gal4* and *ETHR-Gal4* transgenic flies with *UAS-mCD8-GFP* flies to produce progeny expressing GFP in Gal4-expressing cells for immunohistochemical staining. *UAS-mCD8-GFP;ETHR-Gal4>LexAOP-mCherry;GR32A-LexA* were stained for both GFP and RFP. Day 4 adult brains, proboscises, or tarsi were dissected in phosphate buffered saline (PBS) and fixed in 2% paraformaldehyde (Electron Microscopy Sciences Cat# 15713) in PBS for 25 minutes. After washing with PBS-0.2% Triton X-100 (PBST) five times and blocking in 2% normal goat serum (Sigma Cat# G9023) in PBST for 30 minutes at room temperature, tissue samples were incubated with primary antibody; mouse anti-NC82 (DSHB Cat# nc82; RRID:AB_2314866, 1:20) rabbit anti-GFP antiserum (Invitrogen Cat# A10260, 1:1000 dilution), or mouse anti-dsRed (Takara Bio Cat# 632392 1:500 dilution) in PBST for overnight at room temperature. Tissues were washed with PBST five times, and incubated in PBST+2% NGS with secondary; Alexa Fluor 488-labeled goat anti-rabbit IgG (Invitrogen Cat# A27034, 1:500), Alexa Fluor 568-labeled Goat anti-mouse (Biotium, Cat# 20100, 1:500)in room temperature overnight, and washed 4 times for 10 minutes each in PBST before

mounting with Aqua Polymount (Polysciences 18606–20) and imaging. Immunofluorescence was recorded using a confocal microscope (Leica model SP5) with FITC filter in the Institute of Integrative Genome Biology core facility at UC Riverside and processed using the Leica Application Suite X (leica-microsystems.com) or FIJI Imagej (https://fiji.sc).

*Reproductive Tract Staining*: After mating to desired genotype males, *Canton-S* females were flash frozen in liquid nitrogen 2 hrs after mating as in Chen et al., 2022 [65]. Reproductive tracts were removed and fixed 25 minutes in 2% paraformaldehyde, followed by four 10 minute washes in PBST. Tracts were then placed in vectashield mounting medium with DAPI (H-1200) and set overnight, plated and imaged with a Zeiss u880 microscope at either 10X or 40X. Processing was done using FIJI ImageJ (https://fiji.sc).

## Extracellular tip recordings

Extracellular tip-recordings from labellar taste bristles were performed as previously described [66]. All tastants were dissolved in 30 mM tricholine citrate (Sigma, T0252), which functioned as the electrolyte [67]. Recordings were obtained from male flies aged 4–5 days at 25 C. Neuronal responses were amplified and digitized using IDAC-4 data acquisition software. Visualization and manual-quantification of action potential spikes were done with autospike software (Syntech). Number of action potential spikes generated in the first 500 ms following contact artifact was used to quantify neuronal responses.

For recordings with 7-tricosene, solutions were prepared by adding appropriate amounts of 7-tricosene to TCC solutions so as to attain a final TCC concentration of 30 mM right before the day's experiments. Solutions were kept on ice and vortexed right before filling the recording electrodes. Recording electrodes were filled with 7-tricosene solutions by injecting the solutions into them. Electrodes for 7-tricosene application were used only once.

## Injections

Males were injected with saline, 5 pmol ETH (synthesized by Research Genetics), or faux injected (needle penetration without injection of vehicle) using a previously described procedure [12]. In short, males were cold anesthetized and injected abdominally at ZT 16:15 with 0 (faux), 50 nl of saline, or 50 nl of 100μM ETH using a Drummond Nanoject II. After 45 minutes, each group was placed with a virgin female and scored for courtship during a 10 minute interval.

## Methoprene treatment

Methoprene treatment was performed as previously described [12]. Within 4 hours of eclosion, adult males of indicated genotype were chilled on ice in their culture tubes until still and then quickly moved to a cold plate set to 2°C under a dissecting microscope. Using a Nanoject II (Drummond) fit with a blunted capillary males were treated topically on the dorsal side of the abdomen with 50 nl of either ethanol (mutant rescues) or acetone (Inka cell ablation rescues) alone (Fisher MFCD00008765), or 0.01% methoprene (Cayman Chemicals 16807) dissolved in acetone or ethanol (~500 nM). Ethanol was used for mutant testing because acetone treatment slightly increased background courtship, whereas ethanol did not. The entire procedure was completed within 20 minutes, after which flies were returned to their housing.

## Supporting information

**S1 Fig. ETH-deficient males exhibit increased courtship after copulation.** (**A-D**) Cold controls for male-male (**A**, **C**) and male-female (**B**, **D**) courtship indices (time spent courting over

600s) toward a *wt* counterpart for Inka cell-ablated (*ETH-Gal4;Tubulin-Gal80$^{ts}$>UAS-Reaper*, **A-B**), Inka cell secretion-blocked (*ETH-Gal4>UAS-Shi$^{ts}$*, **C-D**), and genetic controls kept in 18˚C after eclosion (ANOVA, n = 15–20). (**E-F**) Post-mating courtship index (time spent courting over total time starting at dismounting, 600s) toward a *wt* female for Inka cell-ablated and genetic control males (**E**, ANOVA, n = 30) and males with EcR knocked down in Inka cells and genetic controls (**F**, ANOVA, n = 10–20). ns p>0.05; *** $p < 0.001$; **** $p < .0001$. (TIF)

**S2 Fig. ETH-deficient males exhibit increased copulation length.** (**A-C**) DAPI-stained female reproductive tracts (Left, scale bar = 250 μm) from *wild-type* females, flash frozen 2 hours after mating to *ETH-Gal4;Tubulin-Gal80$^{ts}$/+* (**A**), *UAS-Reaper/+* (**B**), or *ETH-Gal4; Tubulin-Gal80$^{ts}$/UAS-Reaper* (**C**). Rod-like sperm heads are visible in the spermathecae (right, indicated by yellow arrowheads, scale bar = 25 μm). (**D-E**) Copulation duration (seconds from mounting to dismount) for Inka cell-ablated (*ETH-Gal4;Tubulin-Gal80$^{ts}$>UAS-Reaper*)(**A**, ANOVA, n = 30–40) and Inka cell-blocked (*ETH-Gal4>UAS-Shi$^{ts}$*)(**B**, ANOVA, n = 15–20). ns p>0.05; ** $p < .01$; *** $p < 0.001$. (TIF)

**S3 Fig. Expression pattern of *ETH-Gal4* in adult nervous system.** (**A-B**) Example confocal images of brains (**A-B**) and thoracic ganglia (**C-D**) from male (**A, C**) and female (**B, D**) flies of genotype *UAS-mCD8>ETH-Gal4* (anti-GFP is green, anti-bruchpilot is magenta, scale bars = 50 μm). (TIF)

**S4 Fig. Male expression pattern of *ETHR-Gal4* in taste organs.** (**A-B**) *UAS-GFP;ETHR--Gal4>AOP-mCherry;GR32A-LexA* overlapping expression in labella (**A**, blue arrowheads) non-overlapping expression in tarsi (**B**, black arrowheads). (**C**) Mean spike response from extracellular recordings of S6 taste sensilla of *GR32A-Gal4>UAS-ETHR$^{RNAi1}$* and control males to 10 mM denatonium for 500 ms after contact (One-way ANOVA with Tukey's test for multiple comparison, n = 6–8). (**D**) Spikes/500ms for each 500ms interval between 0 and 10 seconds after contact with 0.5 mM denatonium for GR32A-Gal4>UAS-ETHR-RNAi and genetic controls (One-way ANOVA followed by Tukey's test for multiple comparisons, n = 6–7, color-matched asterisks indicate groups statistically distinct from test flies). (TIF)

## Acknowledgments

We thank Roscoe Huo, Michael Chitgian, Lola Alade, Michelle Gyulnazaryan and Raymond Tan-Tran for scoring courtship videos, Nilay Yapici, Sangsoo Lee, and Yuta Mabuchi for comments on the manuscript, Benjamin White and Naoki Yamanaka for fly strains; Anandasankar Ray for fly lines and helpful advice; and the Bloomington Stock Center (NIH P40OD018537) for reagents. We thank Craig Montell and Nilay Yapici for permitting A.G. and M.M. to complete revision experiments in their respective labs.

## Author Contributions

**Conceptualization:** Matthew R. Meiselman, Michael E. Adams.

**Data curation:** Matthew R. Meiselman, Anindya Ganguly, Michael E. Adams.

**Formal analysis:** Matthew R. Meiselman, Anindya Ganguly, Anupama Dahanukar, Michael E. Adams.

**Funding acquisition:** Michael E. Adams.

**Investigation:** Anindya Ganguly.

**Methodology:** Matthew R. Meiselman, Michael E. Adams.

**Project administration:** Michael E. Adams.

**Resources:** Anupama Dahanukar, Michael E. Adams.

**Software:** Michael E. Adams.

**Supervision:** Anupama Dahanukar, Michael E. Adams.

**Validation:** Matthew R. Meiselman, Anindya Ganguly, Michael E. Adams.

**Visualization:** Matthew R. Meiselman, Michael E. Adams.

**Writing – original draft:** Matthew R. Meiselman, Michael E. Adams.

**Writing – review & editing:** Matthew R. Meiselman, Anindya Ganguly, Anupama Dahanukar, Michael E. Adams.

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
