## [Decision Letter · Decision Letter 0]

31 Mar 2022

Dear Dr Meiselman,

Thank you very much for submitting your Research Article entitled 'Endocrine Modulation of Primary Chemosensory Neurons Regulates  Drosophila Courtship Behavior' to PLOS Genetics.

The manuscript was fully evaluated at the editorial level and by independent peer reviewers. The reviewers appreciated the attention to an important problem, but raised some substantial concerns about the current manuscript. Based on the reviews, we will not be able to accept this version of the manuscript, but we would be willing to review a much-revised version. We cannot, of course, promise publication at that time.

If you decide to revise the manuscript for further consideration at PLOS Genetics, please aim to resubmit within the next 60 days, unless it will take extra time to address the concerns of the reviewers, in which case we would appreciate an expected resubmission date by email to plosgenetics@plos.org.

[LINK]

We are sorry that we cannot be more positive about your manuscript at this stage. Please do not hesitate to contact us if you have any concerns or questions.

Yours sincerely,

Lynn M Riddiford

Associate Editor

PLOS Genetics

Gregory P. Copenhaver

Editor-in-Chief

PLOS Genetics

Your interesting paper requires some revisions before publication as outlined by the 3 reviewers. It is also important that you clearly distinguish between Gr32A and Gr33A receptors, what is known about their roles in both taste and courtship behaviors, and why you chose to use the Gr33A null mutant in Figure 4A, but in most of your studies use Gr32A-expressing neurons. Your present explanation in the paper has not been sufficient as two of the reviewers seem to be confused.

Reviewer's Responses to Questions

**Comments to the Authors:**

Reviewer #1: The manuscript submitted by Meiselman et al is an important addition to the field of Drosophila courtship and endocrinology. The authors carefully characterize the role of the peptide hormone, ETH, in regulating male courtship behavior. Using the Gal4/UAS system to deplete ETH release or ETH receptor activity in certain sensory neurons, the authors link ETH activity to the regulation of male-male courtship and post-copulation courtship toward females. The authors also provide evidence that ETH regulates male courtship indirectly via Juvenile Hormone release from the corpora allata. The authors submit an interesting model whereby ETH influences male courtship through multiple pathways. I recommend this work for publication in PLOS Genetics with the minor revisions enumerated below.

- line 55: cues

- line 68–69: sentence clarification: As it reads, it's not clear if the authors mean that the JH analog rescues the ETH or JH deficiency. Pls specify.

- line 76: What's an "allatotropin"? Pls define it in the text for readers less familiar with insect endocrinology.

- line 78: Readers unfamiliar with the genetic tools that target Inka cells will be curious to know how "clean" the ETH-Gal4 is. Pls clarify. How specific is the ETH-Gal4 to Inka cells in the adult? E.g., can the authors at least rule out CNS expression? In lieu of doing IHC, perhaps the ETH-Gal4 was carefully characterized in another paper; the authors could reference that paper and provide a clear statement about the specificity of the Gal4. I would recommend the same for all Gal4 drivers used in behavioral experiments of the paper. Ideally, the authors would provide adult CNS images for all drivers used in the paper.

- line 86: An ETH rescue experiment would strengthen this section a lot. I recommend that the authors attempt to rescue the phenotype of ETH deficient males by injecting ETH.

- line 92: Do the authors mean Fig. 1F? Also, the authors may consider combining this paragraph with the previous one, as it appears to be a continuation of the remarks re Fig. 1F-G.

- line 118: What is "ETHR-trojan-Gal4" and how exactly does it differ form "ETHR-Gal4" written below? Pls clarify and define in the text.

- line 119: From my understanding of the sentence, the authors are saying that the ETH-trojan-Gal4 labels interneurons that innervate the noted glomeruli, and are post-synpatic to the noted sensory neurons. They claim that these interneurons are also fru positive. Do the authors provide any evidence for that? Pls clarify or provide the data.

- line 121: In Fig. 3A, I recommend including an illustration of a fly brain with the region shown in 3A boxed. At first blush, it's unclear what part of the brain we're looking at in 3A.

- line 135: The authors often say "ETHR-silenced" when referrning to RNAi-mediated depletion. To be more accurate and clear, I recommend "ETHR-depleted" or "ETHR-knockdown".

- line 147: Why did the authors chose to test Gr32A function with a Gr33A mutant? To my knowledge, loss-of-function alleles of Gr32A are available. It would make more sense to use Gr32A mutants. Alternatively, perhaps the authors could use an RNAi against Gr32A.

- line 151: It is indeed surprising that knockdown of ETHR in Gr32A-expressing neurons produced a PCCI phenotype, whereas loss of Gr33A (which should affect the function of Gr32A-expressing neurons) had no apparent effect. Can the authors speculate why this may have happened? Perhaps use of a Gr32A mutant would give different results.

- line 162: The authors should mention was "CA" is.

- line 166–168: This sentence is confusing. The authors write, "restored the normal postmating decline of courtship," but then go onto to say that JH is "insufficient to completely account for elevated post-copulation courtship in ETH-deficient males." As I see it (and what I think the authors are saying), giving JH to ETH-deficient males partially restored the postmating decline of courtship, which suggests that ETH exerts its effects through JH but also directly upon the sensory system.

Along these lines, the authors should state how much methoprene they gave the males in the legend of Fig 4D. Additionally, it is not obvious how we can rule out the possibility that the partial rescue was due to insufficient amounts of methoprene. What if administering >300 nM better rescued the behavior of the ETH-depleted males, possibly even to full levels?

- line 169: In Fig 4D, I recommend that the authors include a wild-type control so the partial rescue can be more apparent to the reader.

- line 403: missing (D)

Reviewer #2: In this interesting paper the authors describe significant new roles for the Ecdysis-Triggering Hormone (ETH). They observe three aspects of courtship that are affected by reduction of ETH or silencing/ablation of cells that produce ETH:

- Injected ETH reduces male-female courtship.

- Ablation of INKA cells, where ETH is produced, increases male-male courtship.

- Silencing of the same cells abolishes post-copulation courtship inhibition.

These deficiencies can be mapped to reduced ETH action in a subset of sensory neurons, and in one case to lower sensitivity to stimuli.

The most significant and novel of these observations is the effect of reduced ETH action on Post-copulatory Courtship Inhibition (PCCI). Mutant males don’t suppress courtship after successful copulation. Based on their examination where ETHR is present, the authors test the roles of several candidate cells. They find that ETHR is required in Gr32 expressing neurons in the labellum and in OR76D antennal glomeruli.

These are really interesting new aspects of the complex physiological adult role of ETH and demonstrate how multi-faceted the adult roles of endocrine regulators can be. The experiments are carefully crafted and done and support the authors’ interpretations well.

I have a few questions and suggestions:

Figure 1:

1. I am a little surprised that mature control animals would only show a 50% copulation rate. Did the authors optimize their assay (and if so how) for this outcome, which would allow them to actually observe an enhancement of copulation success that might otherwise be hidden by a “ceiling effect”?

Figure 2:

The SWITCH experiment is a nice experiments to address the question whether the defect lies in receiver or sender.

1. Have the authors tested a SWITCH constellation where both the first and the second mate are ETH deficient?

Figure 3:

1. The authors show the expression of ETHR-Trojan-Gal4 in distinct antennal lobe neurons. Figure A is not well suited to identify the relevant glomeruli. Without double staining, or at the very least images of the separate channels, outlining and naming the glomeruli (perhaps including a schematic) this is not clear.

2. I believe it would be a good idea to include the S2 fig images in Figure 3.

3. The authors’ finding that Gr32 neurons with ETHR knockdown have reduced sensitivity is intriguing. However, it is not zero. Does the response increase with prolonged exposure or with higher concentration?

4. Why did the authors use Denatonium in their recordings? Why not 7T?

Figure 4:

1.Why was GR33-/- chosen and not GR32-/-?

2. The fact that Gr33-/- did not show a phenotype, but GR32A-Gal4-UAS-ETHR RNAi has not been discussed

3. I think the role of JH deserves a little deeper discussion to describe its roles in different contexts. It appears to be a hormone that is important to “get it right” for wildtype courtship: Normal male-female courtship (enhance courtship when needed), normal postmating inhibition (suppress courtship when it would be inappropriate), perhaps by enabling correct recognition of pheromonal and other cues.

Additonal comments/questions:

1. The authors have previously shown that lower ETH results in reduced courtship memory, but only mention it in passing in the discussion. Are the mutant males described here capable of forming a courtship memory? I would like to see a deeper discussion how reduced courtship memory might play into the current findings.

2. The authors mention that it is unlikely that the absence of PCCI is caused by reduced mating time, in fact it appears slightly longer. Do the mutant males transfer sperm?

Reviewer #3: In their manuscript “Endocrine Modulation of Primary Chemosensory Neurons Regulates Drosophila Courtship Behavior”, Meiselman et al. study the role of Ecdysis Triggering Hormone (ETH) in modulating courtship motivation. For this they look at postcopulatory courtship inhibition (PCCI), the phenomenon of strongly reduced courtship activity of male Drosophila shortly after a copulation. They find that ETH is required for PCCI and the suppression of male-male courtship. Based on a “mate and switch” experiment, they propose that the effect is due to an internal state change of the male, i.e. reduced male detection of courtship inhibitory cues on recently mated females and other males. The authors show that ETH receptor is required in chemosensory neurons Gr32a positive of the labellum for their full sensitivity and inhibition of male-male courtship. ETH receptor knockdown in Gr32a or Or67d sensory neurons also leads to abolishment of the PCCI effect, suggesting that ETH signaling is required for the function of these neurons in sensing inhibitory courtship cues.

The paper addresses an important topic, the neuronal and genetic basis of motivational control of reproductive behaviors. The results are interesting, but main mechanistic questions remain unanswered. Not all results are presented in a clear way and the manuscript and data could be organized in a better way. The Method part is insufficient, not all important experimental detail is explained. Some of the experiments lack critical controls that would allow for an overall more careful, comprehensive and insightful interpretation.

Major points:

1) Inka cell ablation/secretion block: can the authors exclude that killing of Inka cells has other effects than reduction of ETH levels? Do Inka cells possibly secrete other compounds than ETH? Can they specifically affect ETH synthesis in adult Inka cells and confirm their results? The authors inconsistently use “Inka cell ablated” and “ETH deficient” males for denoting the same experimental animals, which is confusing.

2) In Figure 1, temperature controls for the temperature dependent tub-Gal80ts experiments and the Shits experiments are missing, the data is not thus not conclusive.

3) The authors use two assays to assess courtship inhibition toward inappropriate targets: courtship toward mated females (PCCI) and toward males. I am missing a clear discussion how they think these assays differ. Do the authors think that they are testing the same with the two assays, i.e. response to inhibitory courtship cues cVA and 7-T, but PCCI is a more sensitive readout? In the manuscript, differences/commonalities between the assays is not made clear. Thinking a bit more, it is maybe not “surprisingly” (line 151) that PCCI would required both Gr32A and Or67D neurons, whereas male male courtship inhibition is more robust and still in place when only Gr32a is affected.

Fig3D and Fig4A,B belong, in my opinion together and could maybe presented in one figure, also in a more consistent way? For example, why are 2 RNAi lines only used for PCCI, and why is Or47b-Gal4 only tested for male male courtship?

4) Inconsistency of presenting PCCI data in Fig1F and Fig4D versus Fig4A-C. Why is the pre-copulation data not shown in Fig4A-C? Presentation as paired data seems more conclusive to me.

5) Fig 2: When is ETH signaling required in the sensory neurons, during development or in the adult? Does manipulation of ETH signaling for example change sensory neuron survival, wiring/morphology? There are many ways to restrict RNAi knockdown of ETH receptor temporally and by this explore the mechanism of changed response to inhibitory cues.

6) The finding that ETHR knockdown alters the response profile of taste hairs is very interesting. It is however not entirely clear why this is demonstrated with denatonium when the relevant candidate substance is 7-T or extract from male cuticular hydrocarbons. Tests with actual inhibitory courtship cues would be more compelling here. Also, what are the datapoints in Fig3E and 3G representing: tested hairs, animals, repeated stimulus presentations?

7) The last section of the results, line 155 following (Juvenile Hormone regulates post-copulation courtship) is the least clear part of the manuscript. The authors perform methoprene treatment of Inka cell ablated males and find that this partially restores PCCI. However, it is not apparent how JH or its analog methoprene exerts this effect and if sensory neurons are involved. To make the mechanistic link with their previous findings, test their model presented in Figure 4E and disambiguate central motivation circuits from sensory neuron function effects, it would be good to test methoprene treatment in Gr32a and Or67d mutants and flies with ETHR knockdown in sensory neurons.

Minor points:

1) Abstract :«and critical for mate choice and courtship inhibition after the completion of copulation”- the formulation “critical for mate choice after completion of copulation” is a bit confusing/misleading. The term “postcopulatory mate choice” is often used in regards to female decisions about usage of received sperm and control of fertilization. I assume the authors mean the male has a lower courtship motivation right after one copulation and then “chooses” not to mate a second time right afterwards. Since they study courtship motivation rather than “mate choice” sensu stricto, I suggest changing wording here.

2) Line 55: “queues”> cues??

3) Line 140: “ETHR Silencing” > confusing terminology- the receptor is not “silenced”, but its translation diminished by RNAi or it is absent in the mutants.

4) Figure 3A: red green color scheme is inconsiderate for color blind.

5) Line 205: Reference 37 should be rather cited after the following sentence.

**Have all data underlying the figures and results presented in the manuscript been provided?**

Reviewer #1: Yes

Reviewer #2: **No: **No data file has been added yet.

Reviewer #3: **No: **see review, major point 4

PLOS authors have the option to publish the peer review history of their article (what does this mean?). If published, this will include your full peer review and any attached files.

Reviewer #1: No

Reviewer #2: No

Reviewer #3: No

---

## [Editor Report · Decision Letter 1]

27 Jul 2022

Dear Dr Meiselman,

We are pleased to inform you that your manuscript entitled "Endocrine Modulation of Primary Chemosensory Neurons Regulates Drosophila Courtship Behavior" has been editorially accepted for publication in PLOS Genetics. Congratulations!

Yours sincerely,

Lynn M Riddiford

Academic Editor

PLOS Genetics

Gregory P. Copenhaver

Editor-in-Chief

PLOS Genetics

Comments from the reviewers (if applicable):

This revised version is much improved over the original version as you have answered most of the reviewers' comments and criticisms. This paper will make a significant addition to the role of ETH in adult insect reproductive behavior and should be of general interest to all interested in reproductive behavior and how it is coordinated by hormones. When preparing the final version, please make sure that all the references are in the correct format. I notice that the titles of the papers are some times in sentence case as they should be but sometimes have every major word capitalized. Also, genus and species names should be italicized in these titles.

**Data Deposition**

http://datadryad.org/submit?journalID=pgenetics&manu=PGENETICS-D-22-00285R1

**Press Queries**

---

## [Editor Report · Acceptance letter]

18 Aug 2022

PGENETICS-D-22-00285R1 

Endocrine Modulation of Primary Chemosensory Neurons Regulates Drosophila Courtship Behavior 

Dear Dr Meiselman, 

We are pleased to inform you that your manuscript entitled "Endocrine Modulation of Primary Chemosensory Neurons Regulates Drosophila Courtship Behavior" has been formally accepted for publication in PLOS Genetics! Your manuscript is now with our production department and you will be notified of the publication date in due course.

With kind regards,

Zita Barta

PLOS Genetics

On behalf of:
